# Natural reversion promotes LPS elongation in an attenuated *Coxiella burnetii* strain

Carrie M. Long [1] ✉, Paul A. Beare[1], Diane Cockrell[1], Picabo Binette[1], Mahelat Tesfamariam[1], Crystal Richards[1], Matthew Anderson [1], Jessica McCormick-Ell[2], Megan Brose[3], Rebecca Anderson[3], Anders Omsland[4], Talima Pearson [5] & Robert A. Heinzen[1]

Lipopolysaccharide (LPS) phase variation is a critical aspect of virulence in many Gram-negative bacteria. It is of particular importance to *Coxiella burnetii*, the biothreat pathogen that causes Q fever, as in vitro propagation of this organism leads to LPS truncation, which is associated with an attenuated and exempted from select agent status (Nine Mile II, NMII). Here, we demonstrate that NMII was recovered from the spleens of infected guinea pigs. Moreover, these strains exhibit a previously unrecognized form of elongated LPS and display increased virulence in comparison with the initial NMII strain. The reversion of a 3-bp mutation in the gene *cbu0533* directly leads to LPS elongation. To address potential safety concerns, we introduce a modified NMII strain unable to produce elongated LPS.

Lipopolysaccharide (LPS) is a critical component of the outer membrane of Gram-negative bacteria, with functional and structural roles. To date, LPS and the type 4B secretion system (T4BSS) are the only *Coxiella burnetii* virulence factors currently defined by infection of immuno-competent animals[1,2]. Indeed, the virulence potential of *C. burnetii*, the causative agent of human Q fever, in animal models of infection is directly related to LPS length[2]. In vitro propagation of *C. burnetii* leads to LPS phase variation, a process that involves the truncation of phase I, smooth LPS into phase II, rough LPS with lipid A, select core sugars but no *O*-antigen[3–5]. *C. burnetii* LPS phase variation is associated with passage history, as natural isolates first display phase I LPS that shifts to an intermediate, then truncated (phase II) form as early as 10 passages in embryonated hen's eggs[3,4] or, as tested more recently, the synthetic acidified citrate cysteine medium 2 (ACCM-2)[6]. The Nine Mile lineage of *C. burnetii* is commonly utilized for basic experimentation, with Nine Mile I RSA493 (NMI) expressing phase I LPS (fully virulent), Nine Mile Crazy RSA514 (NMC) expressing intermediate, semi-rough LPS (attenuated), and NMII RSA439 (NMII) expressing phase II LPS (avirulent). Moos and Hackstadt demonstrated NMII avirulence by observing a lack

of recovery of NMII from the spleens of intraperitoneally infected (≤10^8 organisms) guinea pigs via passage in embryonated eggs[2]. Further studies identified a large chromosomal deletion in NMII that irreversibly eliminates genes *cbu0678-cbu0698* involved in LPS *O*-antigen biosynthesis[7]. The avirulence of NMII paired with the large chromosomal deletion led to the exemption of NMII, clone 4 as a non-select agent by the US Centers for Disease Control and Prevention's (CDC) Division of Select Agents and Toxins (DSAT) (Fig. S1). A 2018 study by Beare, et al. demonstrated the role of various *C. burnetii* genes in determining LPS length, including *cbu0533*, a predicted undecaprenyl-phosphate alpha-N-glucosamine phosphotransferase enzyme that was associated with LPS truncation in the NMII strain of *C. burnetii*. Here, we describe the natural reversion of *cbu0533* mutation and its impact on LPS length.

## Results

### LPS elongation occurs in *C. burnetii* NMII, clone 4 in vitro and in vivo

To investigate how LPS length influences virulence, guinea pigs were intraperitoneally infected with one of the following Nine Mile *C.*

[1]Laboratory of Bacteriology, Division of Intramural Research, National Institute of Allergy and Infectious Disease, National Institutes of Health, Hamilton, MT 59840, USA. [2]Office of the Director, Office of Research Services, Division of Occupational Health and Safety, National Institutes of Health, Bethesda, MD 20892, USA. [3]Office of the Director, Office of Research Services, Division of Occupational Health and Safety, National Institutes of Health, Hamilton 59840, USA. [4]Paul G. Allen School for Global Health, College of Veterinary Medicine, Washington State University, Pullman, WA 99164, USA. [5]Department of Biological Sciences, Center for Microbial Genetics and Genomics, Northern Arizona University, Flagstaff, AZ 86011, USA. ✉e-mail: carrie.long@nih.gov

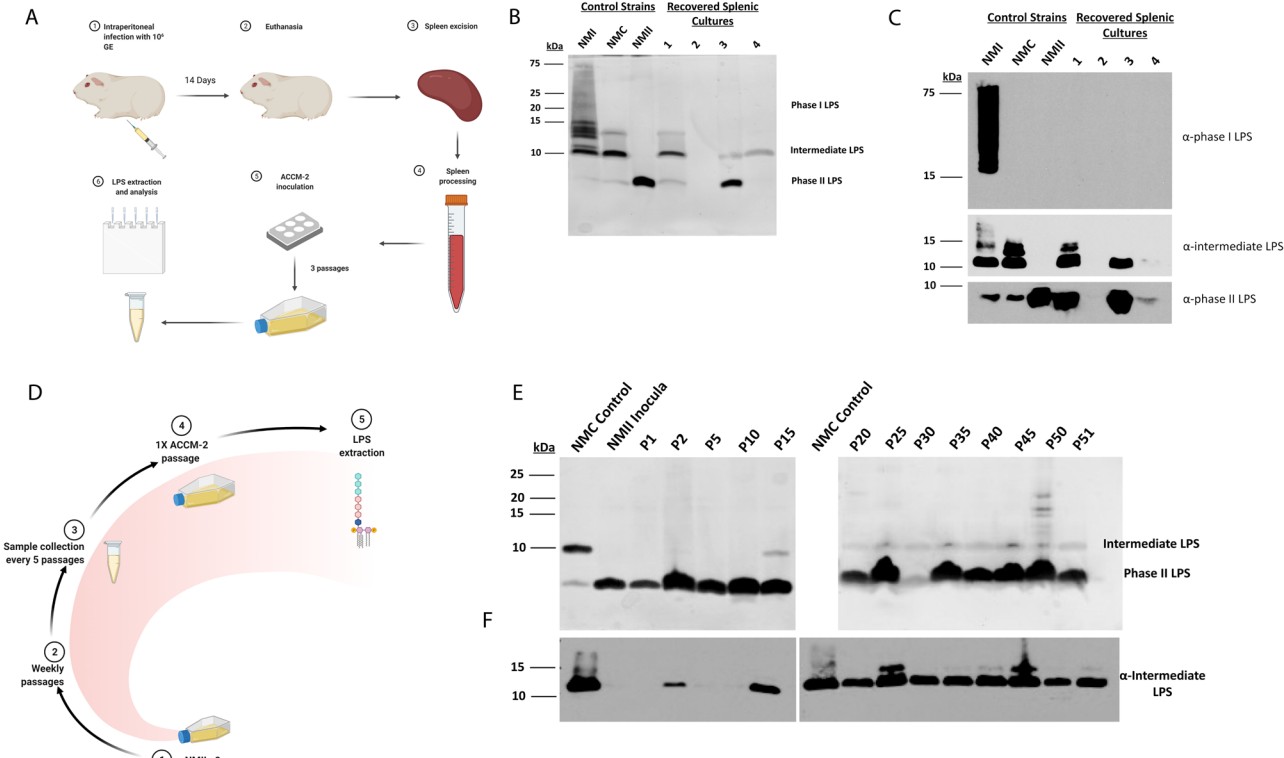

**Fig. 1 | *C. burnetii* LPS elongation occurs in vivo and in vitro.** The in vivo experimental process is described in **A**. Steps 1 and 2 were conducted with ACCM-1 media. Control and recovered splenic isolate LPS from NMII strain infection are depicted following silver stain (**B**) and immunoblot (**C**). Animal group *n* = 4. The in vitro experimental overview is described in **D**. LPS profiles of NMC control, NMII inocula, and ACCM-1 passaged strains following ACCM outgrowth are depicted following silver stain (**E**) and immunoblot with anti-intermediate LPS LPS antibodies (**F**). Images represent single propagated culture samples. Panels **A** and **D** were created with BioRender.com. Source data are provided as a source data file.

*burnetii* strains: NMI, NMC, or NMII (Fig. 1A). The LPS phase of the initial inoculum used in all experiments, as well as from the outgrowth of splenic strains collected at fourteen days post-inoculation, was determined by silver stain (Fig. S2A) and immunoblot (Fig. S2B). NMII starting inocula did not display intermediate LPS banding or specific intermediate LPS reactivity via silver stain or immunoblot, as previously reported[8]. Unexpectedly, strains were recovered from 3 of 4 NMII-infected animals and further analyses revealed intermediate LPS reactivity via silver stain and immunoblot analysis (Fig. 1B, C). We coined this phenomenon "LPS elongation", generating NMII-E (NMII-E1 and NMII-E2 are from two separate splenic cultures). To determine if LPS elongation was limited to in vivo models, long-term axenically passaged NMII was evaluated. NMII was passaged weekly in ACCM-1, with sample collection every 5 passages for 50 weeks (Fig. 1D). LPS analysis revealed the appearance of intermediate LPS reactivity, by passage 15 (silver stain; Fig. 1E) and passage 2 (immunoblot analysis; Fig. 1F). Intermediate LPS reactivity was stable and detectable through the final passage (P51). All described experiments, apart from the initial ACCM-1 long-term study, were performed in BSL-3/ABSL-3 under DSAT select agent guidelines.

### *Cbu0533* mutation reversion causes LPS elongation in *C. burnetii* NMII, clone 4

Previously it has been shown that mutation of *cbu0533*, a gene encoding a protein with homology to undecaprenyl-phosphate alpha-N-glucosamine phosphotransferase enzymes, could be responsible for LPS truncation[9]. To investigate differences in *cbu0533* in NMII-derived strains with elongated LPS compared to NMII, whole genome sequencing (WGS) of splenic-recovered (NMI-E1/2) and ACCM-1 strains was performed. Sequencing showed that *cbu0533* of both virulent NMI and NMC (100% and 99.7% of reads, respectively) encoded a five

leucine stretch beginning at AA168, while NMII unambiguously encoded four leucines, rendering CBU0533 inactive (Table 1). In contrast, two splenic strains exhibiting elongated LPS, NMII-E1 and 2, encoded mixed *cbu0533* leucine profiles, with 62.7% and 10.5% encoding five leucine residues, respectively. In ACCM-1 passaged strains expressing elongated LPS, *cbu0533* exhibited five leucine reads at a 21.1% (passage 26) and 11.9% (passage 51) frequency in contrast to the NMII inocula which encoded four leucine reads. A NMII RSA439 clone 4 Δ*cbu0335* mutant (NMII Δ*0335*) was analyzed due to its extensive passage history in ACCM-2 (>20 passages) and was used to determine if *cbu0533* reversion occurs in ACCM-2-grown bacteria. Like NMII, *cbu0533* sequence reads in the NMII Δ*0335* strain encoded four leucines. In contrast, 99.7% of NMC reads aligned with the NMI sequence encoding five leucine codons. These *cbu0533* sequences corresponded with intermediate and phase II reactivity as determined via silver stain and immunoblot. The presence of large NMII and NMC deletions was confirmed by PCR for all NMII and NMC strains evaluated (Figs. S3 and S4).

### *Cbu0533*-induced LPS elongation enhances the virulence of *C. burnetii* NMII, clone 4

To both characterize the phenomenon of LPS elongation and identify functional consequences, we infected guinea pigs with the following *C. burnetii* strains: NMI, NMC, NMII, recovered splenic strains (NMII-E1 and E2), NMII inocula from the long-term ACCM-1 axenic passaging study (NMII P0), ACCM-1 strain passage 50 (NMII P50), NMI Δ*dot/icm* (a mutated strain unable to replicate intracellularly and cause disease in vivo[1]), and NMII Δ*0533* (a NMII strain lacking the *cbu0533* gene, thus theoretically unable to undergo LPS elongation). Analysis of LPS extracts derived from these inocula revealed expected LPS banding patterns (Fig. S2C, D). NMI exhibited phase I LPS while NMC, NMII-E1,

**Table 1 | Next-generation sequencing analysis of various *C. burnetii* strains following LPS elongation**

| Genome | cbu0533 sequence analysis | | | NMC deletion boundary sequence reads | | | |
|---|---|---|---|---|---|---|---|
| | WT 0533 (LLLLL)[a] #reads | Mutant 0533 (LLLL)[b] #reads | % WT (LLLLL) | 5′ boundary[c] | total reads[d] | 3′ boundary[e] | total reads[f] |
| NMII | 0 | 364 | 0 | 0 | 375 | 0 | 312 |
| NMII Δcbu0335 | 0 | 393 | 0 | 0 | 393 | 0 | 292 |
| NMC | 383 | 1 | 99.7 | 344 | 358 | 312 | 318 |
| NMII P2 ACCM-1 | 0 | 175 | 0 | 0 | 41 | 0 | 80 |
| NMII P26 ACCM-1 | 35 | 131 | 21.1 | 0 | 32 | 0 | 53 |
| NMII P51 ACCM-1 | 37 | 275 | 11.9 | 0 | 30 | 0 | 64 |
| NMII-E1 | 52 | 16 | 62.7 | 0 | 103 | 0 | 69 |
| NMII-E2 | 11 | 75 | 10.5 | 0 | 75 | 0 | 128 |

Note: NMII Δ0335 has >15 ACCM-2 passages. Find all functions in Geneious used for analysis. [a]GCTATTATTATTATTGG sequence; [b]GCTATTATTATTATTGG sequence; [c]ATGCCGACGCCGAG sequence; [d]ATGCCGACGCCG sequence; [e]AGATGCCACCATGT sequence; [f]ATGCCACCATGT sequence.

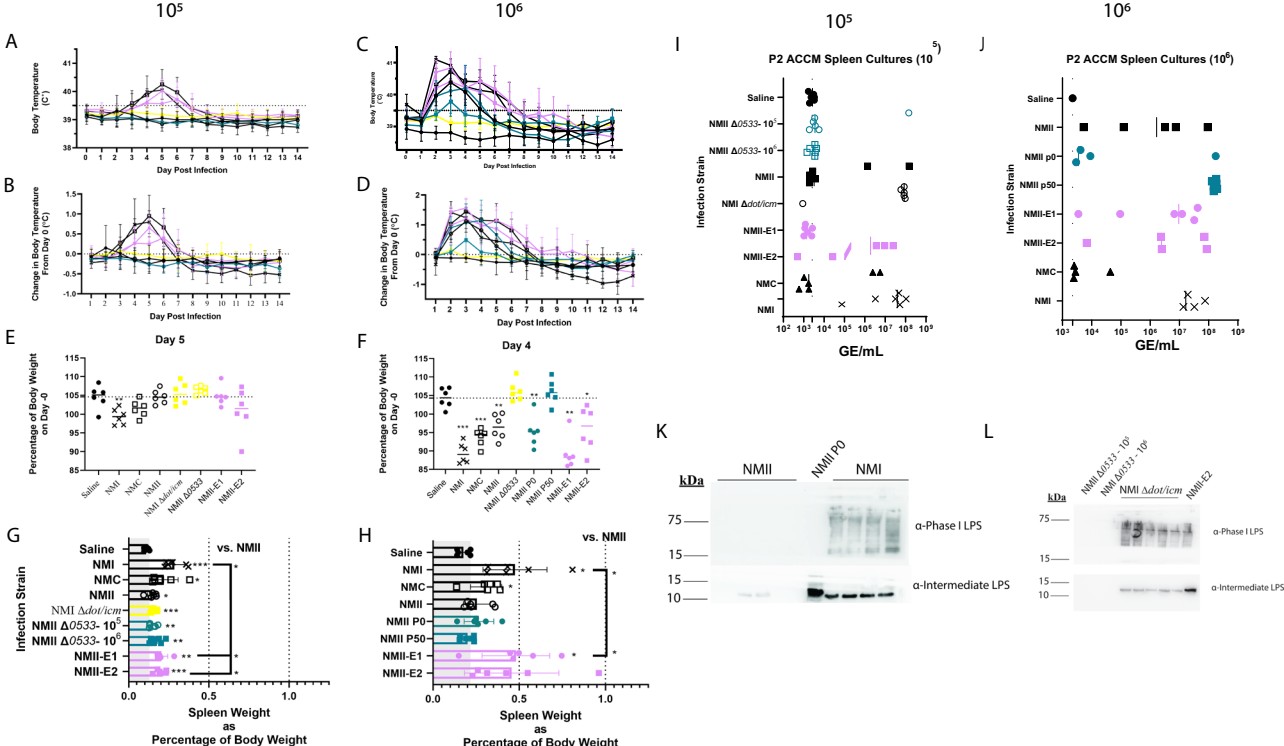

**Fig. 2 | Guinea pig infection reveals increased virulence following LPS elongation.** Guinea pig body temperature (**A**) and change in body temperature (**B**) following a $10^5$ GE *C. burnetii* infectious dose. Guinea pig body temperature (**C**) and change in body temperature (**D**) following a $10^6$ GE *C. burnetii* infectious dose. Fever is indicated by a dotted line at 39.5 °C. Body weight change following a $10^5$ (**E**) or $10^6$ GE (**F**) infectious dose at the time point associated with most severe weight loss in the NMI control group (days 5 and 4, respectively). Black dotted lines indicate the mean percentage in saline control groups. For **A**–**F**, groups are indicated as follows: closed circle:saline, X:NMI, open square:NMC, open circle:NMII, yellow circle: NMI Δ*dot/icm*, yellow circle: NMII Δ*0533*, purple circle:NMII-E1, purple square: NMII-E2. Spleen weight normalized to body weight following a $10^5$ (**G**) or $10^6$ GE (**H**) infectious dose following euthanasia (day 14). Gray blocks represent mean saline group

values. **I** Splenic outgrowth following three ACCM-2 passages as assessed via qPCR following a $10^5$ GE infectious dose. **J** Splenic outgrowth following 3 ACCM-2 passages as assessed via qPCR following a $10^6$ GE infectious dose. **K** LPS profiles of splenic outgrowth following NMII, NMII P0, and NMI infection ($10^5$ GE) as assessed via immunoblot. **L** LPS profiles of splenic outgrowth following NMII Δ*0533*, NMI Δ*dot/icm*, and NMII-E2 infection ($10^6$ GE) as assessed by immunoblot. For **E**–**H**, differences in group means were assessed using two-sample Welch *t*-tests, allowing for unequal variances between groups. For each comparison, we computed Wald-type 95% confidence intervals and described statistical significance with two-sided *p*-values. For all graphs, horizontal bars indicate the group means and error bars represent the group standard deviation. Animal group *n* = 6. * indicates *p* < 0.05 and ** indicates *p* < 0.1. Source data are provided as a source data file.

NMII-E2, and ACCM P50 had intermediate-length LPS. Importantly, NMII and NMII Δ*0533* strains displayed phase II but not intermediate LPS.

Guinea pig body temperatures were monitored for two weeks following inoculation, and NMI, NMC, and NMII-E1 and E2-infected animals experienced fever (39.5 °C) at the low ($10^5$ GE) infectious dose

(Fig. 2A). As expected, neither NMII, NMI Δ*dot/icm*, NMII Δ*0533*-infected, nor saline mock-infected animals experienced fever following inoculation with $10^5$ GE This was also reflected by body temperature change readings normalized to starting body temperature (Fig. 2B). At a higher infective dose ($10^6$ GE), NMII P0, NMII P50, and NMII-infected animals also experienced fever (Fig. 2C). NMII passaged

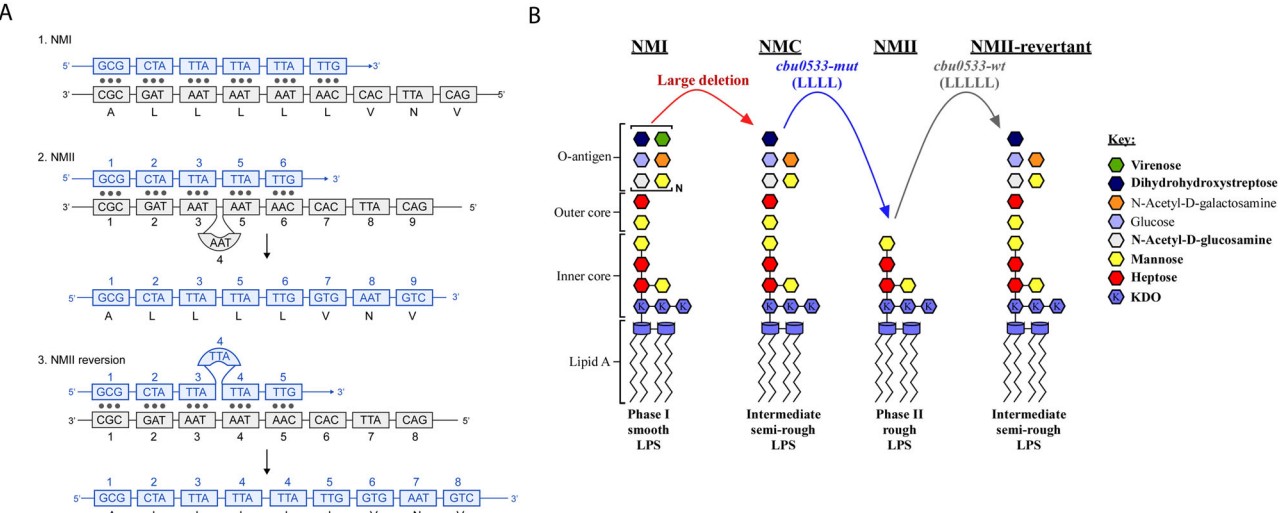

**Fig. 3 | Genetic basis for *cbu0533*-AA168 mutation and impact on *C. burnetii* LPS length.** CbuO533 mutation (likely slipped strand) is depicted in **A**. The impact of *cbu0533* slipped strand mutation and reversion on LPS length is depicted in **B**.

in vitro (P50) caused transient, low-grade fever in a few guinea pigs, while NMII and NMII P0-infected animals experienced fever of intermediate magnitude and duration. NMII *ΔO533*-infected animals did not experience fever (Fig. 2C, D). The inability of NMII *ΔO533* to cause disease is notable, as it is unable to elongate its LPS via reversion of *cbu0533*, directly implicating this gene and its subsequent biological function in restoring virulence to a previously avirulent strain in vivo. In addition to induction of fever, significant weight loss was observed in animals following inoculation with both a high and low dose of NMI (Fig. 2E and Fig. S5A) but was only observed in animals inoculated with NMC, NMII, NMII P0, NMII-E1, or E2 at the high dose (Fig. 2F and Fig. S5B). Similarly, splenomegaly was evidenced in all inoculated animals compared to saline mock-infected animals at the low infection dose, albeit to different magnitudes (Fig. 2G). At the high infective dose, only NMI, NMC, and NMII-E1-inoculated animals exhibited statistically significant splenomegaly compared to saline mock-infected animals (Fig. 2H). Compared to NMI-infected animals, the spleen weights of NMII, NMI Δ*dot/icm*, NMIIΔ*O533*, NMII-E1 and E2, NMII P0, and NMII P50 were significantly decreased. In light of attenuation despite elongated LPS in NMII P50, WGS revealed *cbu0113* mutations that increased in frequency as *C. burnetii* NMII was passaged in ACCM-1 (Table S2). *Cbu0113* encodes a T4BSS effector protein[10,11] that may be responsible for this finding.

To confirm the presence of LPS elongation, bacteria were axenically propagated from spleen homogenates from infected guinea pigs in ACCM-2 and subsequently harvested for LPS extraction and analysis. *C. burnetii* was isolated from individual animals from all groups except for saline mock-infected animals and growth was observed in fourteen of fifty-four cultures in the low-dose experiment and twenty-four of forty-eight cultures in the high-dose experiment (Table S1). GE was quantified for splenic cultures upon ACCM-2 passage and corresponded with the presence of splenic outgrowth for the low (Fig. 2I) and high (Fig. 2J) dose groups. The LPS profiles of recovered bacteria were then examined and NMII, NMII P0, NMII P50, NMII-E1, and NMII-E2 strains exhibited intermediate LPS reactivity while NMI strains exhibited phase I LPS banding and reactivity (Fig. 2K and S6A–C). Although animals infected with NMII Δ*O533* did not display signs of disease, bacteria could be recovered from the spleen; however, NMII Δ*O533* did not display intermediate banding (Fig. S6D) or reactivity (Fig. 2L), consistent with this strain's inability to produce elongated LPS. Similarly, splenic strains of NMI Δ*dot/icm* (high dose) exhibited a mixture of LPS including phase I, intermediate, and phase II banding

pattern detected by silver stain (Fig. S6D) and specific phase I and intermediate LPS reactivity detected by immunoblot (Fig. 2L).

## Discussion

Since the recognition of *C. burnetii* LPS phase variation over half a century ago, this phenomenon has fundamentally impacted Q fever diagnostics and laboratory research. Coincident with the establishment of LPS length as a virulence factor, the search began for genetic lesions that might account for phase variation. Early efforts revealed a missing *HaeIII* fragment of NMII relative to NMI[12] and it was reasoned that this could be related to LPS phase variation[13]. Subsequently, the ~4.4 kb *HaeIII* fragment of NMI was employed in a hybridization scheme to define large and contiguous DNA fragments of ~18 kb and 29 kb missing in NMII and NMC, respectively[14]. Vodkin and Williams further proposed that a point mutation might be responsible for phase II transition in strains lacking large deletions[14]. In 2006, comparative genome hybridization of the NMC and NMII genomes revealed the absence of 20 ORFs (*cbu0679* to *cbu0698*, NMII) and 24 ORFs (*cbu0676* to *cbu700*, NMC), as previously described by Hoover et al.[7,15] and depicted in Fig. S1. Deleted regions were enriched with genes involved in *O*-antigen biosynthesis, including those implicated in the biosynthesis of the sugar virenose[7,16]. Beare, et al. concluded that the large chromosomal deletion explains the intermediate truncation of NMC LPS but does not explain why NMII produces a truncated LPS compared to NMC. Further, the authors posited that NMII likely has an additional point/frameshift mutation, small deletion, or transposon insertion in a gene early in the LPS biosynthetic pathway that was not detected by CGH. In 2017, the NMII genome was sequenced and polymorphisms in several LPS genes were discovered but not discussed in the manuscript[17]. In 2018, Beare et al. used site-directed mutation and complementation to reveal complex, multistep pathways involved in *C. burnetii* LPS phase variation[9]. Our current work reveals a novel determinant of *C. burnetii* phase variation that affects virulence in a guinea pig infection model. This involves *cbu0533* mutation reversion via probable slipped strand mutation, transforming the inactive, NMII gene, to an active gene both in vitro and in vivo (Fig. 3A, B).

Cbu0533 displays homology to *E. coli* WecA (*rfe*) and other bacterial undecaprenyl-phosphate alpha-N-glucosamine phosphotransferases. These enzymes belong to the class of polyisophrenyl-phosphate *N*-acetyl hexosamine-1-phosphate transferases (PNPT) which are integral membrane proteins

found in eukaryotes and prokaryotes[18]. WecA is a common bacterial PNPT that initiates *O*-antigen synthesis in a variety of species;[18] however, *cbu0533* does not complement an *E. coli* Δ*wecA* mutant, suggesting *cbu0533* has *C. burnetii*-specific functionality[19]. Deletion of *cbu0533* in *C. burnetii* results in loss of LPS outer core and O-antigen, suggesting *cbu0533* catalyzes the first step in outer core biosynthesis. Further, leucine 168 appears to be a critical residue essential for the proper function of the enzyme CBU0533 as it is directly downstream of the region homologous to the predicted active site of *E. coli* WecA. The observed LPS elongation in NMII-E strains reported here is characterized by reversion of the *cbu0533* mutation deleting AA168. Importantly, all phase II and intermediate LPS-expressing *C. burnetii* strains analyzed retained large chromosomal deletions of LPS-related genes, indicating that phase II and intermediate LPS-expressing strains with large gene deletions in LPS synthesis machinery cannot produce phase I LPS required for full virulence.

*Cbu0533*-dependent LPS elongation may be related to specific selective pressures. In a guinea pig intraperitoneal infection model, a robust dose of $10^6$ GE may be required as LPS elongation was not detected in animals infected with lower amounts of bacteria. Additionally, LPS elongation has only been observed in ACCM-1 passaged NMII strains. Elevated oxidative stress may provide pressure for LPS elongation, as ACCM-1 is a more oxidative medium than ACCM-D/2[20]. This idea is supported by data indicating that phase II *C. burnetii* are more readily labeled with lactoperoxidase than phase I *C. burnetii*[21]. Here, Hackstadt proposed a model of LPS length-dependent resistance against host clearance pathways such as chemical modification by small molecules including reactive oxygen species. Under certain conditions, short DNA repeats, such as microsatellites, and tandem repeats are susceptible to slipped strand mispairing[22]. Further, bacterial adaptation to host environments via slipped strand mispairing has been documented for several bacterial species, including *Helicobacter pylori*[23,24] and *Neisseria gonorrhoeae*[25]. Mechanisms of phase variation are frequently attributed to stochastic mutations; however, data from a variety of bacteria suggests regulation of these events are influenced by environmental factors[25]. Indeed, host-derived oxidative stress causes double-stranded DNA breaks in *Pseudomonas aeruginosa* genes, leading to mutagenic repair mechanisms and resultant bacterial genetic diversity[26]. The *cbu0533*-AA168 mutation may be a random event followed by biological selection or may be regulated more directly; thus, induced and random mutational events may serve as a benefit to the bacterium leading to enhanced phenotypic diversity and survival in distinct host environments[27].

Beyond *cbu0533*, additional genes appear to influence *C. burnetii* virulence during long-term axenic passage. Kersh, *et al.* observed reduced virulence in a murine infection model following NMI was passaged in ACCM-1[28]. Here, the authors suggested that factors beyond LPS length appear to influence virulence reduction following axenic culture. Our findings bolster this hypothesis, with long-term ACCM-1 passaged strains undergoing LPS elongation with no apparent increase in virulence. WGS revealed *cbu0113* mutations that increased in frequency as *C. burnetii* NMII was passaged in ACCM-1 (Table S2). This gene encodes a T4BSS effector protein[10,11] that may be involved in *C. burnetii* pathogenesis.

NMII has been used in severe combined immunodeficient (SCID) mice for *C. burnetii* virulence factor evaluation[29]. Two studies describing the virulence of NMI and NMII in SCID mouse models yielded contrasting findings. Andoh, et al. reported low virulence of NMII[30] while Islam, et al. described high virulence of this strain[31]. Both studies employed the same intraperitoneal infection model and doses of the respective strains; however, the experimental timing was distinct. These disparate results prompt inquiry regarding the potential occurrence of LPS reversion in SCID mice. Further investigation is required to elucidate potential effects on virulence in this context due

to LPS reversion. This approach could decipher host factors influencing LPS reversion. Indeed, additional model systems such as mice, *Drosophila melanogaster*[32], and *Galleria mellonella*[33] should be examined for LPS elongation.

The importance of LPS in *C. burnetii* biology and Q fever pathogenesis is demonstrated not only by reduced virulence of NMII but also by the *cbu0533*-dependent, virulence-enhancing effect of LPS elongation in NMII-E strains. Due to *C. burnetii*'s high infectivity, aerosol transmission potential, and a history of causing laboratory-acquired infections[34], *C. burnetii* strains are considered US CDC DSAT select agents and biosafety level 3 (BSL-3) organisms. NMII RSA439, clone 4 is the sole select agent-exempt strain of *C. burnetii* (https://www.selectagents.gov/exclusions-hhs.html); this is based on the strain's clonal nature, avirulence, and historic lack of splenic recovery in a guinea pig infection model, and non-revertible nature due to the large chromosomal deletion[35]. Here, we demonstrate splenic recovery of *C. burnetii* following infection with NMC or NMII at considerably lower infection doses ($10^{5-6}$ GE). The discrepancy between historic experiments and this study is likely due to variations in the culturing technique, as previously utilized egg yolk cultures may be less sensitive and successful at isolating bacteria from infected tissues compared to optimized axenic growth media. Here, we present the *C. burnetii* LPS elongation in vitro and in vivo. Our data identify *cbu0533* as the gene mediating phase II to intermediate LPS transition and the ensuing increase in virulence potential. The *cbu0533* mutation occurring in NMII is characterized by the loss of a leucine-encoding fragment and is likely a slipped strand event. LPS elongation in NMII-E represents mutation reversion to the leucine-encoding region normally lost in wild-type. The resultant impact on LPS length harbors profound implications for laboratory work with *C. burnetii*, diagnostic sensitivity, and vaccine development. Also, *cbu0533* reversion and LPS elongation impart increased virulence potential on NMII strains derived from guinea pigs. Although this prompts regulatory and safety concerns, virulence reestablishment is at an intermediate level. We have observed LPS elongation after two passages in ACCM-1 axenic media and a single infection passage in guinea pigs. Importantly, the guinea pig is considered the most physiologically relevant small animal model of human Q fever due to a dose-fever response similar to that of humans[36]. Thus, the findings presented here raise concerns regarding the impact of LPS elongation on humans and in additional model systems such as frequently utilized animal model systems and cell culture. As demonstrated here, the reversion of *cbu0533* mutation associated with increased virulence may be specific to high-dose guinea pig infections which would justify the exemption status of NMII in the context of cellular culture systems. Although NMII has a decades-long history of safe use at BSL-2 containment we believe it is incumbent that individual laboratories assess whether *cbu0533* reversion has occurred in their specific laboratory applications. Our construction of NMII Δ*0533* provides an alternative strain that is non-revertible and biologically equivalent to NMII for general laboratory purposes, allowing for the control of LPS elongation and associated virulence enhancement.

## Methods

### Ethics

All experiments were performed in accordance with the RML Institutional Biosafety Committee and DSAT regulations. Animals were housed in approved animal biosafety level 3 (ABSL-3) facilities and manipulated under ABSL-3 standard operating procedures approved by the Rocky Mountain Laboratories Institutional Biosafety Committee and an Institutional Animal Care and Use Committee-approved protocol (2021-067-E). Animal experiments and procedures were performed in an Association for Assessment and Accreditation of Laboratory Animal Care-accredited National Institutes of Health-National Institute of Allergy and Infectious Diseases animal facility.

## *Coxiella burnetii* strains and propagation

All *C. burnetii* strains were grown in acidified citrate cysteine medium-1, -2 or -D (ACCM-1, ACCM-2, or ACCM-D)[6] at 37 °C, 2.5% $O_2$, and 5% $CO_2$ and were subsequently stored at −80 °C in cell freezing medium (DMEM with 10% fetal bovine serum and 10% dimethyl sulfoxide). The *cbu0335* and *cbu0533* genes were deleted in NMII RSA439, clone 4 using ACCM-2 and ACCM-D axenic media, respectively, as previously described (*2*). For experiments based on long-term axenic passaging, ACCM-1 was inoculated with *C. burnetii* NMII RSSA439, clone 4, and passaged weekly. Samples were then expanded with a single passage in ACCM-2 media and LPS was extracted as described below. For all experiments, wild-type NMII was derived from a Vero cell infected *C. burnetii* NMII RSA439 strain harvested at 28 days post-inoculation. Oligonucleotide primers and plasmids used in this study are listed in Supplementary Table 3.

## Biosafety and dual-use research of concern oversight

All manipulations of phase I and intermediate LPS-expressing *C. burnetii* and infected animal tissue were performed in a BSL-3 laboratory following standard operating procedures approved by the Rocky Mountain Laboratories Institutional Biosafety Committee. *C. burnetii* expressing phase I and elongated LPS were handled as select agents as defined by the US Division of Select Agents and Toxins (DSAT). These samples were classified as select agents and were manipulated and stored accordingly. Specifically, axenic NMII RSA439, clone 4 manipulations were performed in BSL-2 containment, and all animal infection studies and downstream analysis were performed in ABSL-3/BSL-3 containment. The work described in this report was evaluated per the HHS policies for dual-use research of concern (DURC) and potential pandemic pathogens (P3CO). The NIH Dual Research of Concern review entity reviewed and approved the project and this manuscript before publication. The committee determined that this research did not meet the definition of DURC because the results, information, and technology are not reasonably anticipated to be directly misapplied to have a significant impact on public health.

## Characterization of *C. burnetii* LPS and relative splenic bacterial load

*C. burnetii* LPS was extracted by modified hot phenol extraction and visualized by silver stain and immunoblot[9]. All samples utilized in silver stains and immunoblots were derived from the indicated experiments and were run in parallel. The BioRad Precision Plus Protein Dual Color Standard was utilized as a molecular weight marker in these experiments. *C. burnetii* stocks used for infection were quantified using qPCR to enumerate genome equivalents (GE)[6]. For enumeration of relative bacterial load, spleens were homogenized and homogenates were used to inoculate ACCM-2 media as previously described[1], with the following modifications. Briefly, 200 uL of spleen homogenate was added to 1.8 mL of ACCM-2 in 6-well plates. After 7 days of growth, *C. burnetii* DNA was extracted from bacteria contained in 1.4 mL of each ACCM-2 culture using the Qiagen DNeasy Blood and Tissue Kit. Quantification of genome equivalents (GE) in extracted DNA was conducted by qPCR using primers specific for *C. burnetii groEL*. Ct values over 35 were considered background amplification in this assay.

## Guinea pigs

Female Hartley guinea pigs were obtained from Charles River (strain code 051) at 4 to 6 weeks of age. Female animals were utilized to control for known impacts of sex on *C. burnetii* pathogenesis/virulence and in accordance with historical studies. Animals were acclimated for at least a week before experimental manipulation. To minimize potentially confounding sex-associated factors, female guinea pigs were utilized in these studies. Animals were housed in individually ventilated plastic cages (Allentown; two animals per cage) with hardwood Sani-chip bedding (PJ Murphy). A high-fiber guinea pig diet (Envigo global high-fiber guinea pig diet; Teklad, cat n. 2041) and chlorinated, reverse osmosis filtered tap water were administered ad libitum. A 12-h light–dark cycle was maintained in animal housing facilities which were kept at 68–72 °F and 40–60% relative humidity with a 50% set point. Experimental group numbers ranged from 4 to 6 animals and are described in corresponding figure legends. All animals were housed in approved animal biosafety level 3 (ABSL-3) facilities and manipulated under ABSL-3 standard operating procedures approved by the Rocky Mountain Laboratories Institutional Biosafety Committee and an Institutional Animal Care and Use Committee-approved protocol. Animal experiments and procedures were performed in an Association for Assessment and Accreditation of Laboratory Animal Care-accredited National Institutes of Health-National Institute of Allergy and Infectious Diseases animal facility.

## *C. burnetii* infection, euthanasia, tissue collection, and processing

On the day of infection, animals were sedated by isoflurane inhalation using an anesthetic vaporizer with activated charcoal adsorption filters (VetEquip Inc, cat. n. 901801 and 931401) and an IPTT-300 transponder (BioMedic Data Systems) was implanted subcutaneously above the shoulder of each animal in a longitudinal orientation using a large bore needle. Guinea pigs were infected with $10^{5-6}$ GE of *C. burnetii* in USP-grade saline via intraperitoneal injection. Negative control animals were mock infected with USP-grade saline for each experiment. Body weights, body temperatures, and behavioral/clinical changes were recorded daily following infections. Body temperatures were collected using a DAS-8007-P reader (Bio-Medic Data Systems) at a consistent daily time. A temperature of ≥39.5 °C was defined as fever[1]. For all experiments, guinea pigs were humanely euthanized on day 14 post-inoculation. Blood was collected by cardiac puncture using Vacutainer® blood collection tubes and needles (BD). Following euthanasia, spleens were excised and placed into conical tubes containing 5 mL sterile phosphate-buffered saline (PBS; Gibco, pH: 7.4, cat. no. 10010023). Spleens were dissociated using disposable 15- or 50-mL tissue grinders (VWR International, cat. n. 47732-446 and -450). Spleen cellularity was determined using a Scepter automated cell counter (Millipore, cat. n. PHCC20060) with size exclusion parameters (6 to 36 µm). Spleen suspensions were stored at −20 °C for subsequent analysis.

## Splenic bacterial culture and LPS analysis

Bacteria isolated from spleens were expanded in ACCM-2 as indicated above. For initial cultures, 200 uL of spleen homogenate was added to 1.8 mL of ACCM-2 in 6-well plates. After 7 days of growth, cultures were passaged 1/10 into new 6-well plates. After an additional 7 days of growth, cultures were passed into T25 flasks (1 mL of passage 2 culture into 6 mL ACCM-2) and propagated for an additional 7 days. DNA was extracted from bacteria contained in 1.4 mL of each ACCM-2 culture using the Qiagen DNeasy Blood and Tissue Kit and qPCR was conducted as described above. LPS was extracted and analyzed by silver stain and immunoblot as above.

## PCR and next-generation sequence analysis of *C. burnetii* strains

Genomic DNA (gDNA) was isolated from *C. burnetii* strains using the PowerMicrobial Maxi DNA isolation kit (Mo Bio) or Qiagen DNeasy Blood and Tissue kit with an additional step of boiling for 30 minutes before the physical disruption of the cells. Genomic DNA was sequenced using an Illumina MiSeq instrument to generate read pairs as previously described[17,37]. PCR was performed on isolated gDNA with Accuprime Pfx or Taq (Invitrogen) using the oligonucleotide pairs listed in Supplementary Table 3. Genetic analysis was performed using Geneious software.

## Statistical analysis

Statistical analyses were conducted using GraphPad Prism version 7.0 (GraphPad Software, La Jolla, CA, USA). Statistical evidence for differences in group means was assessed using two-sample Welch $t$-tests, allowing for unequal variances between groups. For each comparison, we computed Wald-type 95% confidence intervals and described statistical significance with two-sided $p$-values, where we represent $p$-values as equal to or below 0.05 with a single asterisk (*) and $p$-values equal to or below 0.01 with a double asterisk (**). For splenic bacterial outgrowth data, we utilized the same analysis after transforming all samples above our qPCR amplification threshold ($Ct > 35$) to the minimum threshold value. For each comparison, we describe statistical significance with two-sided $p$-values, where we represent $p$-values as equal to or below 0.05 with a single asterisk (*), $p$-values equal to or below 0.01 with a double asterisk (**), and $p$-values equal to or below 0.001 with a triple asterisk (***).

## Reporting summary

Further information on research design is available in the Nature Portfolio Reporting Summary linked to this article.

## Data availability

The sequencing data generated in this study have been deposited in the Genbank database under the accession codes found in Table S3. Source data are provided in this paper.

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

## Acknowledgements
We thank Rose Perry-Gottschalk (NIH/NIAID/DIR/Visual Medical Arts) and BioRender for assistance with figure design. We thank the Rocky Mountain Veterinary Branch for animal caretaking. We thank Ryan Richards (NIH/NIAID/DIR) for assistance in editing the manuscript. Biorender was utilized for figure creation. This work was supported by the Intramural Research Program of the National Institutes of Health, National Institute of Allergy and Infectious Disease (1ZIAAI001331, Long).

## Author contributions
Conceptualization: CML, PAB, RAH; methodology: CML, PAB, DC, PB, MT, CR, MA, AO, TP; investigation: CML, PAB, DC, PB, MT, CR, MA, AO, TP; visualization: CML, PAB, PB, MT, CR, MA, AO, TP; funding acquisition: CML, RAH; project administration: CML, JME, MB, RA; supervision: CLM, RAH; writing—original draft: CML, RAH; writing—review and editing: CML, PAB, DC, PB, MT, CR, MA, JME, MB, RA, AO, TP, RAH.

## Funding

## Competing interests
The authors declare no competing interests.
