## [Peer Review File · Nature Communications]

Natural reversion promotes LPS elongation in an attenuated *Coxiella burnetii* strainReviewer #1 (Remarks to the Author):

This report provided the first evidence to suggest that the Cbu0533 gene may be responsible for *C. burnetii* NMII LPS elongation and avirulent NMII strain demonstrated the potential to be able to reverse into a more virulent NMC strain with an intermediated LPS in both in vivo and in vitro systems. This study is significant and important, particularly, NMII strain is considered as a biosafety level 2 organism, exempted from the CDC select agent program and has been used extensively in studying host-pathogen interactions and immunogenicity in vitro and in vivo systems in BSL2 laboratories worldwide. In addition, the experiments are well designed and conducted by an outstanding research team.

The observation that a more virulent NMII strain (NMII-E) with a similar intermediated LPS profile as the NMC strain was recovered from a single NMII infection in guinea pigs for a short time frame is interesting and surprising. However, due to the large (26 kb) chromosomal deletion of LPS biosynthetic genes in NMII organism plus the obligate intracellular lifestyle and slow growth nature of *C. burnetii*, it has been considered that NMII organism is unable to revert into a virulent *C. burnetii* strain via natural genetic transformation of deleted LPS biosynthetic genes. In addition, despite NMII strain has been used extensively in studying host-pathogen interactions and immunogenicity in vitro and in vivo systems since 1956, there is currently no evidence to suggest that NMII strain will regain *C. burnetii* virulent LPS production capacity by transduction and conjugation processes. Since NMII-E was recovered from a single NMII infection of guinea pigs for a short time frame (14 days post-infection) and the bacterial recovering process in vitro was taken 3 passages in ACCM-2 culture for 21 days, it is unclear if the NMII LPS elongation occurred during the infection in guinea pigs or during the in vitro recovering process that needed for passaging and culture in ACCM-2. Thus, additional evidence that can establish a direct linkage between NMII and NMII-E would provide strong support to the conclusion of such significant study. Using a GFP-express NMII strain that do not carry NMC specific genes in guinea pig infection and in vitro culture systems might be help to establish a direct linkage between NMII and NMII-E.

Specific comments:

1. The purity of *C. burnetii* strains was evaluated by silver staining and western blotting, which may be not sensitive enough to exclude the possibility of a small numbers of NMC bacteria presence in NMII inocula for the both guinea pig and in vitro experiments. Use PCR to test whether NMC specific genes can be amplified from NMII inocula would be helpful.
2. The passage history in ACCM culture system for NMII strain is unclear. The mutation rate seems high in NMII strain that used for this study. Is there any speculations about why mutants occurred in Cbu0533 gene of the NMII strain? How the mutations in other genes of NMII strain?
3. It would be interesting to see if the LPS structure and sugar components are identical between NMII-E and NMC strains by mass spectrometry analysis.
4. This might be beyond the scope of this report but it would be interesting to see how deletion of Cbu0533 gene in NMI and NMC strain would affect the LPS profiles of NMI and NMC strains.
5. Using the SCID mouse model to confirm the observations would provide additional support.

Reviewer #2 (Remarks to the Author):

Review

Manuscript # NCOMMS-23-34462-T

Natural reversion promoting LPS elongation of an attenuated *Coxiella burnetii* strain

The current manuscript addresses a question that has long existed in the *Coxiella burnetii* (Cb) research field relative to the organisms LPS length; its first recognized virulence determinant. It has long been appreciated that the Cb Nine Mile II (NMII) strain with truncated LPS demonstrated a loss of virulence in immunocompetent animal models, while the Cb Nine Mile "Crazy" (NMC) strain, with intermediate length LPS, demonstrate an intermediate virulence. However, early

genomic studies between the Cb Nine Mile I (NMI) wild-type strain, the fully truncated NMII, and the NMC intermediate strain clearly did not explain the phenomenon since the NMII strain had a smaller genomic deletion (26kb) than the NMC strain (31.6kb), and that the NMII deletion fell within the NMC deletion. This clearly suggested that other genetic changes were responsible for the NMII LPS truncation, leading to a loss of virulence. The authors appear to have found the gene at the crux of this question; Cb cbu0533. More specifically, they identified an AA168 deletion in NMII that is not present in NMI or NMC. The authors used a combination of Guinea pig infections and splenic recovery, serial axenic media passage, high throughput DNA sequence analysis of recovered Cb, genome and LPS length comparisons, followed by analysis of cbu0533 deletion mutants and reversion isolates to establish and confirm their findings in vivo. Their creation of a Cb NMII cbu0533 that appears incapable of the LPS partial reversion and subsequent intermediate virulence provides an alternative Cb strain if studies or lab settings require.

The manuscript is concise and well written. The studies are straightforward, well designed and thoroughly performed using robust indicators of Cb growth, genomics, virulence measures in animal models, and LPS structure detection. While axenic media has greatly enhanced the ability to grow and isolate Cb in the lab, it remains a tedious, meticulous and challenging procedure, whether performed in BSL2 or BSL3 settings. The same applies for work with Cb in an animal BSL3 environment.

Some issues need to be addressed to avoid confusion and provide more clarity within the manuscript and supplemental material. They are described below:

1. RSA Strain numbers used: There are a few places where RSA 439 is called NMI and RSA 493 is called NMII. While it's certainly easy to have a typo or paste issue with these strains, it's crucial they be clear in this manuscript particularly given its content and subject matter. Introductory paragraph lines 46-49. Supplemental material MM-Biosafety paragraph indicates that "NMI RSA 439 clone 4" manipulations were performed in a BSL2 setting. This is likely a simple typographical error.
2. Table S3 – There appears to be 2 forward primers indicated for NMC/NMII-deldet. Again, likely a typographical error in the second oligo.
3. Fig S3/Table S3 – a sentence or two on where the oligo primers are located within the Cb genome as well as the PCR amplification products that should, or should not, appear depending on the genome template would be helpful. Since the "NMII-deldet" and "NMC/NMII-deldet" oligo sequences exist in NMI, this explanation would make the experiment more readily clear to readers.
4. Given Cb NMI virulence in IP Guinea pig models, it would be useful for the authors to address the use of 100,000 and 1,000,000 Cb genome equivalents as inoculums and possibly speculate on even lower inoculum numbers relative to Cb NMII and Cb NMI-E strains. While the authors did discuss the differences in recovery of Cb from spleens following 10^5 and 10^6 GE inoculation, it would be interesting to consider lower infectious dose as it relates to reversion and laboratory use.

In summary, the current study is of particular significance to the field of *Coxiella burnetii* basic and diagnostic research while having implications for other facultative and obligate intracellular pathogens. It offers the opportunity to reexamine what virulence means relative to this organism and the possibility of some partial dose-dependent virulence increase associated with a genetic reversion in an established non-virulent strain.

Reviewer #3 (Remarks to the Author):

In this study by Long et al., the authors describe a method for generating a *C. burnetii* NMII mutant with increased virulence. Serial passage of NMII in vitro and bacteria recovered from infected guinea pigs revealed a 3 bp mutation in cbu0533 that resulted in elongation of LPS. The study is well designed is greatly significant for both researchers in this study and the regulatory consideration of Select Agents.

Major

The authors demonstrate reversion of cbu0355 in vitro following axenic passage in ACCM-1. However, it is unclear whether this phenomenon is restricted to ACCM-1 or occurs in NMII propagated in ACCM-2/-D, which are more commonly used in the field. Can the authors comment on whether they also evaluated the effects of passage in ACCM-2 or -D and if so whether LPS elongation also occurred?

Some concerns over Fig 1E and F. Are these images representative of several cultures or just one? The authors claim to detect intermediate LPS by passage 2 (line 73; Fig 1F). However, confirmation by silver stain does not occur until passage 15. Can the authors provide an explanation as to why a) silver stain is negative until later passage, and b) why the immunoblot is negative for intermediate LPS at passage 5 and 10? NMII P2 ACCM1 has 0% WT (LLLLL) or WT cbu0355 reads (table 1), also suggesting LPS elongation does not occur this early. P30 is negative for silver stain and positive for immunoblot without explanation.

The authors describe similar virulence between NMII and NMII P0 in vivo at a higher dose. Are there any differences between these two inocula in the way they are cultured? What was the reason for omitting NMII P0 and NMII P50 from the low dose challenge? Can the authors briefly describe the differences between NMII-E1 and -E2. Are these two separate cultures from a single guinea pig that has been infected with NMII, or splenic cultures from two separately infected animals?

Minor

Line 46 – 48: "Nine Mile I RSA439 (NMI) expressing phase I LPS (fully virulent), Nine Mile Crazy RSA514 (NMC) expressing intermediate, semi-rough LPS (attenuated), and NMII RSA493 (NMII)". Edit so that NMI is RSA493 and NMII is RSA439

Line 450: Presumably "*** indicates $p > 0.1$ ", should read "*** indicates $p < 0.01$ "?

Fig 1A and 1D are low resolution, difficult to read text

Fig 2 is low resolution. Difficult to read some of the data, cannot see "grey blocks" referenced in the legend, axis for Fig 2I and J are unreadable.

Check formatting of references

Reviewer #4 (Remarks to the Author):

In the study presented, Long et al further characterise the lipopolysaccharide content of several model isolates of *Coxiella burnetii*, a bacterial pathogen which, with its full length LPS, is considered a biothreat agent that requires enhanced biosecurity measures under select agent guidelines. The authors could show that avirulent nature of the LPS deficient isolate NMII, which is classed as a non-select agent, might not be as stable as previously thought, since their result from guinea pig infections describe a so far unreported phenomenon of reversion of a rough (phase II) LPS phenotype to a more virulent intermediate LPS phenotype.

These results have a profound impact on the Coxiella community, who has used the NMII isolate at BSL-2 containment for many years, and so will the described mutant which could be the solution for this security issue.

Some comments require addressing though:

1) The authors use the term "isolate" for both the guinea pig and passaged samples. The Materials & Methods section does not mention any isolation on solid media to single colony level, and the SNP data presented in Table 1 also suggests the presence of a mixed population in these samples. Therefore, the term isolate is misleading and should be changed to samples or cultures or similar.

2) The authors claim that LPS elongation is achieved by slipped strand reversion in cbu0533. Despite this being the most likely explanation, it should be noted that this is the proposed model only (e.g. in the Legend to Fig 3 and in sentence in lines 170-171 and 243). The legend for Fig. 3 A) could also do with more of an explanation.

3) Previous results on cbu0533 (described in reference #9), including the work using complemented / active site mutants, should be described in more detail for context, perhaps in the introduction, as well as the fact that the leucine repeat length variation in NMII is not a novel finding of this study.

4) The Materials & Methods section describes that the guinea pig samples were repeatedly passaged in ACCM-2 medium, presumably in order to increase bacterial yields. In the discussion section (lines 189-190), the authors mention that "LPS elongation has only been observed in ACCM-1 passaged NMII isolates". It is important to point out to the reader that passage in ACCM-2 does not result in LPS elongation, as otherwise the LPS elongation observed in the guinea pig samples might have occurred during the in vitro passages.

5) Statistical analysis in Fig. 2 should be performed by comparing the evolved samples to NMII, not NMI; similarly, data for the cbu0533 mutant at 10^6 GE should be included in the 10^6 graphs, not (just) in the 10^5 graphs, i.e. in panels F, H, and J.

6) The use of the Δ cbu0335 mutant (lines 89-92 and Table 1) needs further explanation. Why was this mutant and not the NMII wt passaged in ACCM-2 included in the data?

7) It is unclear whether the changes in cbu0533 are the only mutations observed in the whole genome sequenced samples that show LPS elongation. If other SNPs were present, these should be shown.

8) Extended Table 2 should be moved from the discussion to the results section in a new paragraph where effects of other genetic determinants on virulence and the difference between the seemingly less virulent ACCM-P50 sample and the more virulent E1/2 guinea pig samples is addressed.

Minor comments:

1) Why is the phase I LPS of the NMI control not visible in Fig S5 A and C? Are these also post infection guinea pig samples?

2) Figures 2 K&L and Fig S3&S5: it should be made clear what the samples are, e.g. are all lanes labelled NMII the reference strain or do these lanes actually contain the guinea pig and passaged samples too? In the latter case, please consider naming the samples as done in Fig 1 and Fig S2

3) The introductory paragraph is missing references, e.g. the paper by Moos and Hackstadt is mentioned again in line 48, but the reference number (presumably #2) is missing, as well as a reference for the statement in lines 51-52.

4) Table S1: NMI Δ 0dot/icm: should this be : NMI Δ dot/icm?

5) Table S2: *C. burnetii* should be in italics

6) Carefully check the references e.g. for species names in italics or left over html code such as  etc

Reviewer #1 (Remarks to the Author):

This report provided the first evidence to suggest that the Cbu0533 gene may be responsible for *C. burnetii* NMII LPS elongation and avirulent NMII strain demonstrated the potential to be able to reverse into a more virulent NMC strain with an intermediated LPS in both in vivo and in vitro systems. This study is significant and important, particularly, NMII strain is considered as a biosafety level 2 organism, exempted from the CDC select agent program and has been used extensively in studying host-pathogen interactions and immunogenicity in vitro and in vivo systems in BSL2 laboratories worldwide. In addition, the experiments are well designed and conducted by an outstanding research team.

The observation that a more virulent NMII strain (NMII-E) with a similar intermediated LPS profile as the NMC strain was recovered from a single NMII infection in guinea pigs for a short time frame is interesting and surprising. However, due to the large (26 kb) chromosomal deletion of LPS biosynthetic genes in NMII organism plus the obligate intracellular lifestyle and slow growth nature of *C. burnetii*, it has been considered that NMII organism is unable to revert into a virulent *C. burnetii* strain via natural genetic transformation of deleted LPS biosynthetic genes. In addition, despite NMII strain has been used extensively in studying host-pathogen interactions and immunogenicity in vitro and in vivo systems since 1956, there is currently no evidence to suggest that NMII strain will regain *C. burnetii* virulent LPS production capacity by transduction and conjugation processes. Since NMII-E was recovered from a single NMII infection of guinea pigs for a short time frame (14 days post-infection) and the bacterial recovering process in vitro was taken 3 passages in ACCM-2 culture for 21 days, it is unclear if the NMII LPS elongation occurred during the infection in guinea pigs or during the in vitro recovering process that needed for passaging and culture in ACCM-2. Thus, additional evidence that can establish a direct linkage between NMII and NMII-E would provide strong support to the conclusion of such significant study. Using a GFP-express NMII strain that do not carry NMC specific genes in guinea pig infection and in vitro culture systems might be help to establish a direct linkage between NMII and NMII-E.

We thank the reviewer for their positive comments and thorough analysis. Based on the following data, we feel that NMII LPS elongation occurs during guinea pig infection: The single in vitro passages were conducted in ACCM-2 media following guinea pig infection; although ACCM-1 passage appears to induce LPS elongation, we have not observed LPS elongation occurring in ACCM-2 passaged strains, as demonstrated by the *C. burnetii* *cbu0355*^{-/-} strain, which didn't experience LPS elongation after > 20 passages in ACCM-2. In reference to the NMC gene analysis, please refer to the response to comment 1 below.

Specific comments:

1. The purity of *C. burnetii* strains was evaluated by silver staining and western blotting, which may be not sensitive enough to exclude the possibility of a small numbers of NMC bacteria presence in NMII inocula for the both guinea pig and in vitro experiments. Use PCR to test whether NMC specific genes can be amplified from NMII inocula would be helpful.

We agree that gene-specific qPCR is critical for high resolution analysis of NMC vs NMII identification. We performed this assay with NMII and NMC strains (Supplemental Figure 3) and found that both NMII and NMC had the large "NMI deletion" but only NMC contained the "NMC deletion". The sequencing data from each strain was screened for sequences specific to the large NMC deletion. These sequences

were only found in the NMC strain. Thus, this shows that NMII strains do not express the full NMC deletion.

2. The passage history in ACCM culture system for NMII strain is unclear. The mutation rate seems high in NMII strain that used for this study. Is there any speculations about why mutants occurred in Cbu0533 gene of the NMII strain? How the mutations in other genes of NMII strain?

We agree that this information is important and have added it to the “*Coxiella burnetii* strains and propagation” section of the Materials and Methods.

We speculate that the *cbu0533* mutation reversion occurs due to host selective pressure and enhanced bacterial fitness with elongated LPS in specific stress states. The *cbu0533* mutation also occurs in long term passage of NMC (Beare, et al., doi: 10.1371/journal.ppat.1006922) but we have not observed *cbu0533* mutation/reversion in other *C. burnetii* strains, indicating that this mutation may be specific to Nine Mile strains; however, this has not been firmly established. As demonstrated by Beare, et al. (doi: 10.1371/journal.ppat.1006922), other *C. burnetii* strains can develop diverse LPS-related mutations that impact LPS structure, such as the Australia strains.

3. It would be interesting to see if the LPS structure and sugar components are identical between NMII-E and NMC strains by mass spectrometry analysis.

We agree and have attempted to determine the LPS structure of *C. burnetii* yet we have experienced setbacks with insufficient quantity and purity. We continue to attempt these experiments. Because the monoclonal NMC antibody utilized in LPS Western blotting reacts with both NMII-E and NMC, we speculate that there is a high degree of structural similarity between the LPS content of these two strains. As documented by Beare, et al. (doi: 10.1371/journal.ppat.1006922), wild-type Australia strain RSA297 does not react with NMII-specific antibody via Western blot, further supporting the resolution of this method for distinguishing variable LPS structure.

4. This might be beyond the scope of this report but it would be interesting to see how deletion of Cbu0533 gene in NMI and NMC strain would affect the LPS profiles of NMI and NMC strains.

This has been previously performed (Beare, et al., doi: 10.1371/journal.ppat.1006922) with NMI and after removing *cbu0533*, this strain was unable to produce NMI or NMC reactivity and is identical to the NMII LPS profile, in line with our conclusions here.

5. Using the SCID mouse model to confirm the observations would provide additional support.

We agree and mention this from lines 215-222. Future studies are planned to address this inquiry.

Reviewer #2 (Remarks to the Author):

Review

Manuscript # NCOMMS-23-34462-T

Natural reversion promoting LPS elongation of an attenuated *Coxiella burnetii* strain

The current manuscript addresses a question that has long existed in the *Coxiella burnetii* (Cb) research field relative to the organisms LPS length; its first recognized virulence determinant. It has long been appreciated that the Cb Nine Mile II (NMII) strain with truncated LPS demonstrated a loss of virulence in immunocompetent animal models, while the Cb Nine Mile “Crazy” (NMC) strain, with intermediate length LPS, demonstrate an intermediate virulence. However, early genomic studies between the Cb Nine Mile I (NMI) wild-type strain, the fully truncated NMII, and the NMC intermediate strain clearly did not explain the phenomenon since the NMII strain had a smaller genomic deletion (26kb) than the NMC strain (31.6kb), and that the NMII deletion fell within the NMC deletion. This clearly suggested that other genetic changes were responsible for the NMII LPS truncation, leading to a loss of virulence. The authors appear to have found the gene at the crux of this question; Cb cbu0533. More specifically, they identified an AA168 deletion in NMII that is not present in NMI or NMC. The authors used a combination of Guinea pig infections and splenic recovery, serial axenic media passage, high throughput DNA sequence analysis of recovered Cb, genome and LPS length comparisons, followed by analysis of cbu0533 deletion mutants and reversion isolates to establish and confirm their findings *in vivo*. Their creation of a Cb NMII cbu0533 that appears incapable of the LPS partial reversion and subsequent intermediate virulence provides an alternative Cb strain if studies or lab settings require.

The manuscript is concise and well written. The studies are straightforward, well designed and thoroughly performed using robust indicators of Cb growth, genomics, virulence measures in animal models, and LPS structure detection. While axenic media has greatly enhanced the ability to grow and isolate Cb in the lab, it remains a tedious, meticulous and challenging procedure, whether performed in BSL2 or BSL3 settings. The same applies for work with Cb in an animal BSL3 environment.

We thank the reviewers for their thorough analysis and positive comments.

Some issues need to be addressed to avoid confusion and provide more clarity within the manuscript and supplemental material. They are described below:

1. RSA Strain numbers used: There are a few places where RSA 439 is called NMI and RSA 493 is called NMII. While it's certainly easy to have a typo or paste issue with these strains, its crucial they be clear in this manuscript particularly given its content and subject matter. Introductory paragraph lines 46-49. Supplemental material MM-Biosafety paragraph indicates that “NMI RSA 439 clone 4” manipulations were performed in a BSL2 setting. This is likely a simple typographical error.

This has been corrected (lines 46, 48, materials and methods).

2. Table S3 – There appears to be 2 forward primers indicated for NMC/NMII-deldet. Again, likely a typographical error in the second oligo.

This has been corrected.

3. Fig S3/Table S3 – a sentence or two on where the oligo primers are located within the Cb genome as well as the PCR amplification products that should, or should not, appear depending on the genome template would be helpful. Since the “NMII-deldet” and “NMC/NMII-deldet” oligo sequences exist in NMI, this explanation would make the experiment more readily clear to readers.

We agree and have created a figure visually outlining the location of the primers (Figure S4).

4. Given Cb NMI virulence in IP Guinea pig models, it would be useful for the authors to address the use of 100,000 and 1,000,000 Cb genome equivalents as inoculums and possibly speculate on even lower inoculum numbers relative to Cb NMII and Cb NMI-E strains. While the authors did discuss the differences in recovery of Cb from spleens following 10^5 and 10^6 GE inoculation, it would be interesting to consider lower infectious dose as it relates to reversion and laboratory use.

We selected the 10^6 GE doses based on prior studies and the lack of evidence for LPS elongation at a dose of 10^5 in guinea pigs when archived spleen samples from these studies were evaluated. Thus, we posit that at doses below 10^5 reversion would not occur or are below our limit for detection. We believe that we captured the interface between LPS elongation occurring (10^6) and not occurring (10^5).

In summary, the current study is of particular significance to the field of *Coxiella burnetii* basic and diagnostic research while having implications for other facultative and obligate intracellular pathogens. It offers the opportunity to reexamine what virulence means relative to this organism and the possibility of some partial dose-dependent virulence increase associated with a genetic reversion in an established non-virulent strain.

We agree and thank the reviewer again for their careful consideration of the manuscript.

Reviewer #3 (Remarks to the Author):

In this study by Long et al., the authors describe a method for generating a *C. burnetii* NMII mutant with increased virulence. Serial passage of NMII in vitro and bacteria recovered from infected guinea pigs revealed a 3 bp mutation in *cbu0533* that resulted in elongation of LPS. The study is well designed and is greatly significant for both researchers in this study and the regulatory consideration of Select Agents.

We thank the reviewer for their positive comments and thorough review of the manuscript.

Major

The authors demonstrate reversion of *cbu0355* in vitro following axenic passage in ACCM-1. However, it is unclear whether this phenomenon is restricted to ACCM-1 or occurs in NMII propagated in ACCM-2/-D, which are more commonly used in the field. Can the authors comment on whether they also evaluated the effects of passage in ACCM-2 or -D and if so whether LPS elongation also occurred?

Based on the following data, we feel that NMII LPS elongation occurs exclusively in ACCM-1 culture conditions: The single in vitro passages were conducted in ACCM-2 media following guinea pig infection; although ACCM-1 passage appears to induce LPS elongation, we have not observed LPS elongation occurring in ACCM-2 passaged strains, as demonstrated by the *C. burnetii cbu0355*^{-/-} strain, which didn't experience LPS elongation after > 20 passages in ACCM-2. Further, we have analyzed RNAseq data of other NMII mutants passaged greater than 20 times in ACCM-2 or ACCM-D and did not detect the *cbu0533* mutation reversion.

Some concerns over Fig 1E and F. Are these images representative of several cultures or just one? The authors claim to detect intermediate LPS by passage 2 (line 73; Fig 1F). However, confirmation by silver stain does not occur until passage 15. Can the authors provide an explanation as to why a) silver stain is negative until later passage, and b) why the immunoblot is negative for intermediate LPS at passage 5 and 10? NMII P2 ACCM1 has 0% WT (LLLLL) or WT *cbu0355* reads (table 1), also suggesting LPS elongation does not occur this early. P30 is negative for silver stain and positive for immunoblot without explanation.

These images are representative of a single culture. This has been denoted at line 405.

Intermediate LPS reactivity occurs by passage 2, as indicated by the banding pattern described in Figure 1F. The ratio of NMC to NMII at P2 is very low, it is possible that we just don't have the read depth to detect the presence of the *cbu0533* mutation.

- a) We speculate that silver stain detection of LPS is less specific compared to Western blotting; thus, reactivity may not have been detected given the lower amount.
- b) Although the intermediate LPS bands are fainter than the surrounding samples, it appears that P5 and P10 samples are reacting with intermediate LPS.

The authors describe similar virulence between NMII and NMII P0 in vivo at a higher dose. Are there any

differences between these two inocula in the way they are cultured? What was the reason for omitting NMII P0 and NMII P50 from the low dose challenge?

All NMII inocula was initially isolated from a Vero cell infection (day 28 harvest), this information has been added to the “*Coxiella burnetii* strains and propagation” section of the materials and methods. Beyond number of experimental ACCM-1 passages, these strains were all derived from NMII-P0.

Due to feasibility (ABSL-3 experiments), we had to limit numbers of guinea pigs in each experiment; thus, we chose to omit NMII P0 and P50 samples at the 10^5 GE dose. This was decided based on the lack of LPS elongation observed in NMII P0 prior to guinea pig infection and the presence of a *cbu0113* (T4SS substrate) mutation in the P50 strain, likely reducing the virulence of this strain.

Can the authors briefly describe the differences between NMII-E1 and -E2. Are these two separate cultures from a single guinea pig that has been infected with NMII, or splenic cultures from two separately infected animals?

NMII-E1 and NMII-E2 are splenic cultures from two separately infected animals. This has been clarified in the text (lines 69-70)

Minor

Line 46 – 48: “Nine Mile I RSA439 (NMI) expressing phase I LPS (fully virulent), Nine Mile Crazy RSA514 (NMC) expressing intermediate, semi-rough LPS (attenuated), and NMII RSA493 (NMII)”.

Edit so that NMI is RSA493 and NMII is RSA439

This has been corrected.

Line 450: Presumably “** indicates $p > 0.1$ ”, should read “** indicates $p < 0.01$ ”?

This has been corrected.

Fig 1A and 1D are low resolution, difficult to read text

This has been corrected.

Fig 2 is low resolution. Difficult to read some of the data, cannot see “grey blocks” referenced in the legend, axis for Fig 2I and J are unreadable.

We have increased the axis font size and the overall size of the I and J graphs to improve readability.

Check formatting of references

This has been corrected (Lines 311 and 338).

Reviewer #4 (Remarks to the Author):

In the study presented, Long et al further characterise the lipopolysaccharide content of several model isolates of *Coxiella burnetii*, a bacterial pathogen which, with its full length LPS, is considered a biothreat agent that requires enhanced biosecurity measures under select agent guidelines. The authors could show that avirulent nature of the LPS deficient isolate NMII, which is classed as a non-select agent, might not be as stable as previously thought, since their result from guinea pig infections describe a so far unreported phenomenon of reversion of a rough (phase II) LPS phenotype to a more virulent intermediate LPS phenotype.

These results have a profound impact on the Coxiella community, who has used the NMII isolate at BSL-2 containment for many years, and so will the described mutant which could be the solution for this security issue.

We appreciate the reviewer's thoughtful evaluation.

Some comments require addressing though:

1) The authors use the term "isolate" for both the guinea pig and passaged samples. The Materials & Methods section does not mention any isolation on solid media to single colony level, and the SNP data presented in Table 1 also suggests the presence of a mixed population in these samples. Therefore, the term isolate is misleading and should be changed to samples or cultures or similar.

We agree and have changed this to strain (lines 62, 64, 67, 83, 86, 139, 140, 144, 191, and 220).

2) The authors claim that LPS elongation is achieved by slipped strand reversion in cbu0533. Despite this being the most likely explanation, it should be noted that this is the proposed model only (e.g. in the Legend to Fig 3 and in sentence in lines 170-171 and 243). The legend for Fig. 3 A) could also do with more of an explanation.

We have softened our language regarding the representation of this conclusion (lines 171, 245, and 472-474).

3) Previous results on cbu0533 (described in reference #9), including the work using complemented / active site mutants, should be described in more detail for context, perhaps in the introduction, as well as the fact that the leucine repeat length variation in NMII is not a novel finding of this study.

We have included this information in further detail in the introduction (lines 55-58).

4) The Materials & Methods section describes that the guinea pig samples were repeatedly passaged in ACCM-2 medium, presumably in order to increase bacterial yields. In the discussion section (lines 189-190), the authors mention that "LPS elongation has only been observed in ACCM-1 passaged NMII isolates". It is important to point out to the reader that passage in ACCM-2 does not result in LPS

elongation, as otherwise the LPS elongation observed in the guinea pig samples might have occurred during the in vitro passages.

Based on the following data, we feel that NMII LPS elongation occurs during guinea pig infection: The single in vitro passages were conducted in ACCM-2 media following guinea pig infection; although ACCM-1 passage appears to induce LPS elongation, we have not observed LPS elongation occurring in ACCM-2 passaged strains, as demonstrated by the *C. burnetii* *cbu0355*^{-/-} strain, which didn't experience LPS elongation after > 20 passages in ACCM-2.

5) Statistical analysis in Fig. 2 should be performed by comparing the evolved samples to NMII, not NMI; similarly, data for the *cbu0533* mutant at 10⁶ GE should be included in the 10⁶ graphs, not (just) in the 10⁵ graphs, i.e. in panels F, H, and J.

We have performed statistical analysis comparing the evolved samples to NMII instead of NMI and have adjusted the graph accordingly.

We chose to group the *cbu0533* mutants together, regardless of dose, as these groups were included in the same independent experiment as the additional strains at 10⁵ GE. To facilitate proper statistical analysis, we compare experimental to control groups within the same experiment.

6) The use of the Δ *cbu0335* mutant (lines 89-92 and Table 1) needs further explanation. Why was this mutant and not the NMII wt passaged in ACCM-2 included in the data?

The initial explanation of the use of *C. burnetii* Δ 0335 is located from lines 94-97. We chose this strain as a passage control as it already had a large number (>20) of ACCM-2 passages.

7) It is unclear whether the changes in *cbu0533* are the only mutations observed in the whole genome sequenced samples that show LPS elongation. If other SNPs were present, these should be shown.

This is an important consideration. The whole genome sequences for NMII-E1 and NMII-E2 can be found in our deposited data within Genbank. Less than 8 additional mutations were observed in these strains, with three being conserved within wild-type NMII. Further, none of these genes are related to LPS functionally, indicating that the *cbu0533* mutation reversion is the driver for LPS elongation in NMII.

8) Extended Table 2 should be moved from the discussion to the results section in a new paragraph where effects of other genetic determinants on virulence and the difference between the seemingly less virulent ACCM-P50 sample and the more virulent E1/2 guinea pig samples is addressed.

This text has been moved to the results section (lines 135-137).

Minor comments:

1) Why is the phase I LPS of the NMI control not visible in Fig S5 A and C? Are these also post infection guinea pig samples?

The lack of strong NMI banding via silver stain may be due to LPS truncation occurring in the NMI control due to ACCM passage and/or the low resolution of silver stains.

These samples are post infection guinea pig samples, with the exception of the NMI control, which was included for comparative analysis.

2) Figures 2 K&L and Fig S3&S5: it should be made clear what the samples are, e.g. are all lanes labelled NMII the reference strain or do these lanes actually contain the guinea pig and passaged samples too? In the latter case, please consider naming the samples as done in Fig 1 and Fig S2

All samples are from guinea pig splenic samples and are labeled according to the inocula the guinea pigs were exposed to.

3) The introductory paragraph is missing references, e.g. the paper by Moos and Hackstadt is mentioned again in line 48, but the reference number (presumably #2) is missing, as well as a reference for the statement in lines 51-52.

This has been corrected.

4) Table S1: NMI $\Delta 0\text{dot}/\text{icm}$: should this be : NMI $\Delta\text{dot}/\text{icm}$?

This has been corrected.

5) Table S2: *C. burnetii* should be in italics

This has been corrected.

6) Carefully check the references e.g. for species names in italics or left over html code such as *etc*

This has been corrected.

Reviewer #1 (Remarks to the Author):

Thank you for the responses to all my concerns. All the questions have been addressed.

Reviewer #2 (Remarks to the Author):

Nature Communications
Manuscript # NCOMMS-23-34462-T

Natural reversion promoting LPS elongation of an attenuated *Coxiella burnetii* strain

The current manuscript addresses a question that has long existed in the *Coxiella burnetii* (Cb) research field relative to the organisms LPS length; its first recognized virulence determinant. It has long been appreciated that the Cb Nine Mile II (NMII) strain with truncated LPS demonstrated a loss of virulence in immunocompetent animal models, while the Cb Nine Mile "Crazy" (NMC) strain, with intermediate length LPS, demonstrate an intermediate virulence. However, early genomic studies between the Cb Nine Mile I (NMI) wild-type strain, the fully truncated NMII, and the NMC intermediate strain clearly did not explain the phenomenon since the NMII strain had a smaller genomic deletion (26kb) than the NMC strain (31.6kb), and that the NMII deletion fell within the NMC deletion. This clearly suggested that other genetic changes were responsible for the NMII LPS truncation, leading to a loss of virulence. The authors appear to have found the gene at the crux of this question; Cb cbu0533. More specifically, they identified an AA168 deletion in NMII that is not present in NMI or NMC. The authors used a combination of Guinea pig infections and splenic recovery, serial axenic media passage, high throughput DNA sequence analysis of recovered Cb, genome and LPS length comparisons, followed by analysis of cbu0533 deletion mutants and reversion isolates to establish and confirm their findings in vivo. Their creation of a Cb NMII cbu0533 that appears incapable of the LPS partial reversion and subsequent intermediate virulence provides an alternative Cb strain if studies or lab settings require.

The manuscript is concise and well written. The studies are straightforward, well designed and thoroughly performed using robust indicators of Cb growth, genomics, virulence measures in animal models, and LPS structure detection. While axenic media has greatly enhanced the ability to grow and isolate Cb in the lab, it remains a tedious, meticulous and challenging procedure, whether performed in BSL2 or BSL3 settings. The same applies for work with Cb in an animal BSL3 environment.

The authors addressed the concerns that were indicated for this reviewer.

The study is of particular significance to the field of *Coxiella burnetii* basic and diagnostic research while having implications for other facultative and obligate intracellular pathogens.

Reviewer #3 (Remarks to the Author):

Comments adequately addressed by author

Reviewer #4 (Remarks to the Author):

The authors have addressed all of my previous concerns in their amended version of the manuscript.